# Closing Connectivity Gap: An Overview of Mobile Coverage Solutions for Not-Spots in Rural Zones

**DOI:** 10.3390/s21238037

**Published:** 2021-12-01

**Authors:** Diego Fernando Cabrera-Castellanos, Alejandro Aragón-Zavala, Gerardo Castañón-Ávila

**Affiliations:** 1School of Engineering and Science, Tecnologico de Monterrey, Ave. Epigmenio González 500, Querétaro 76130, Mexico; aaragon@tec.mx; 2School of Engineering and Science, Tecnologico de Monterrey, Ave. Eugenio Garza Sada 2501, Monterrey 64849, Mexico; gerardo.castanon@tec.mx

**Keywords:** aerial communication, FANET, not-spots, stratospheric communication platform, UAV, UAV-assisted network, 5G

## Abstract

Access to broadband communications in different parts of the world has become a priority for some governments and regulatory authorities around the world in recent years. Building new digital roads and pursuing a connected society includes looking for easier access to the internet. In general, not all areas where people congregate are fully covered, especially in rural zones, thus restricting access to data communications and inducing inequality. In the present review article, we have comprehensively surveyed the use of three platforms to deliver broadband services to such remote and low-income areas, and they are proposed as follows: unmanned aerial vehicles (UAV), altitude platforms (AP), and low-Earth orbit (LEO) satellites. These novel strategies support the connected and accessible world hypothesis. Hence, UAVs are considered a noteworthy solution since their efficient maneuverability can solve rural coverage issues or not-spots.

## 1. Introduction

Coverage indicators are essential for perceiving the reliability of the network in a determined area. Specifically, each country defines the best practices to determine the covered zones for their boundaries and, therefore, the appropriated thresholds associated with frequency bands. Commonly, most mobile operators offer coverage on the main urban area [1], limiting the countryside to lower bandwidth, thus reducing connection speeds [2]. Nevertheless, the interest in providing more connectivity in rural zones has grown in the last decade since economic development will be an immediate fact.

Extensive terminology has arisen to address the coverage holes, wherein a few or even any operator guarantee its services. The Ofcom—Office of Communication of the United Kingdom—names them as *Not-Spots*. The prior entity has the intention to reach the coverage index until 95% by 2022 [3]. Several British operators (O_2_, Vodafone, EE, and Three) have implemented a sharing strategy, allowing a mutual infrastructure approach and, therefore, improving the competition in the countryside. This layout—or National Roaming—grants customers in rural areas the possibility of connecting to the strongest available signal, regardless of the chosen operator for these clients [3].

### 1.1. Motivation

The inequality in accessing Information and Communications Technologies (ICT) resources and the lack of opportunities for reaching development are the most significant drawbacks in developing countries, even though mobile devices accounted for 87% of broadband connections there [4]. Latin America is not so far from that situation. However, most governments have changed their ways of supporting more connectivity opportunities in the last decade.

Within the call for promoting a prosperous society, which can curb inequality and poverty, the United Nations (UN) has considered the access to fixed-broadband internet under the Goal 9 outline—Industry, Innovation, and Infrastructure—a valuable resource for population growth. By 2018, 96.5% of the entire world population can access at least 2G mobile networks where LTE covers 81.8% of the population [5]. In full swing of the Internet Era, not all villages can leverage granted-by-connectivity opportunities because of thef high cost of access, which unearths the at-risk population group’s unfairness.

Considering the ongoing demands of communication infrastructure, the UN Sustainable Goal 9 aims to significantly increase ICT resources access by 2020, in addition to struggling to hook up LDC (Least Developed Countries) with affordable technology [6]. The COVID-19 pandemic has triggered comprehensive research and investment in digitalization, namely economy and education boosting, since teleworking, video conferencing systems, and remote education have been crucial parts during pandemic and post-pandemic times.

In order to assess the connectivity situation around the world, the GSM Association (GSMA) provides the GSMA Mobile Connectivity Index (MCI), which measures the performance of 170 countries based on four key enablers of mobile internet adoption—infrastructure, affordability, consumer readiness, and content and services—where the current data include 2019 [7]. The prior institution has released The State of Mobile Internet Connectivity 2020 Report, which analyzes the critical connectivity trends from 2014 to 2019 in terms of mobile internet use [4].

The coverage has not been sufficiently wide to provide the same standards compared to Europe. For instance, in [4], it is possible to check that Europe and Central Asia and North America were more 70% connected than compared to 54% in Latin America and the Caribbean. Despite these observations, it is crucial to note that the offered services have grown in the last region since its MCI overcomes a 61 score by 2019 in contrast with that obtained five years earlier: 51 [7].

Although MCI appears to be the most significant, this is not the only affair to highlight at the moment in terms of analyzing connectivity for particular contexts, such as the countryside. Therefore, it is necessary to map out the earlier metric with each country’s rural population density, discovering the most important limitations that prevent people from adopting mobile internet. Table 1 depicts both MCI and Rural Population Density (RPD) —in percentage units from the total—in order to analyze the gap among fifteen Latin countries.

Identifying the locations where coverage is under specific boundaries appears to be suitable for sketching out the Not-Spots’ presence in these contexts. Therefore, Figure 1 charts the correlation between the total population and coverage density, segregated by mobile network generation from 2G to novel 5G, in two specific countries of the target region: Colombia (Figure 1a) and Mexico (Figure 1b). This is aimed at recognizing the coverage gap inside the mentioned countries. Moreover, it likely identifies regions in which the population cannot access voice and data services.

### 1.2. Paper Outline

We have reviewed several strategies that pursue new connectivity standards by expanding network coverage, especially for developing countries, compared with developed countries such as European countries. These approaches aim to list the possible technologies that will improve connectivity in rural zones after studying the researched options in the alternative deployment of networks for optimizing those regions.

After stating the motivations and the principal purposes roughly, we outlined the article as follows: Section 2 presents a perspective of the network environment (outdoor and indoor), highlighting the solutions that engage emergent services such as the Internet of Things (IoT). Section 3 sketches out the possible researched technologies in order to enhance coverage in rural zones and achieve high Quality of Service (QoS) and the network’s throughput. Section 4 and Section 5 set forth the discussion and conclusions about the assessed solutions in prior sections, in addition to bearing in mind new research opportunities in this field. To this end, Figure 2 shows the overall organization of the cited references in this review article.

## 2. The Rural Paradigm Shift

Under the perspective of granting better connectivity standards in the countryside, it is adequate to set forth the differences among several best-fitted technologies in order to find an optimal solution. The first approach is a suitable onset to focus on mobile network connections optimized for rural populations and self-steady links for IoT terminals, whether involving new communication tendencies such as Device-to-Device communications (D2D) or even 5G [8].

Outdoor and indoor environments require the above aims to lift specific responses within rural population needs. The outdoor schemes consider current traffic estimation of the mobile network by algorithmic focusing since it may provide a proper breakdown for determining the cells’ coverage capacity [9]. With respect to indoor environments, achieving an extended coverage based on ad hoc Networks by lower frequency bands involving repeaters would be suitable [10]. Nonetheless, other approaches will be analyzed in Section 2.1 and Section 2.2 that cope with issues for both cases.

### 2.1. Outdoor Perspective

Gatwaza et al. in [9] highlighted that traffic is an outstanding factor to dimension current mobile networks. In isolated zones, the challenge lies in finding out how to fix the maximum coverage per single base station with respect to complex topography and highly dispersed population distribution [11]. Information on geographical distribution is quite relevant for internet service providers (ISP) since it allows the estimation of areas that deserve specialized deployment toward determining the under-requirements of system capacity [12].

The coverage parameter defines the network’s scope, resulting in the expected enhancement for lower-connectivity regions. Consequently, the channel’s propagation parameters, such as Path Loss Exponent and Losses, are essential for coverage and quality analysis. For instance, CDMA and AMPS cells may overlay the targeted geographical areas to carry information among remote Base Stations (BS) appropriately [13]. Other alternatives include the use of the WiMAX—IEEE 802.16—set of standards [14] and TV White Spaces (TVWS) [15] to enable a ubiquitous network.

At the onset of 21st century, the developed countries evaluated options to achieve better QoS in rural zones. One of them was implementing high-quality in-car mobile services without the implementation of new cell sites. Thus, there was a possibility to raise roadway coverage areas by using antenna arrays set over constant on-way cars. This advance might have allowed the minimization of cost with respect to the non-installation of more BSs. Furthermore, it would provide improvements due to its implementation over dynamic CDMA signals, eradicating AMPS services [13].

With the massification of novel technologies, e.g., 5G and IoT for urban zones, the idea includes analyzing other low-deployment cost options, such as FTTx. Araujo et al. pointed out in [16] that services on FTTC (Curb) would be 70% cheaper than 5G implementation and 20% less expensive than FTTH (Home). Although the main idea is boosting countrysides as potential high opportunity zones, not all operators expect to invest in high-cost infrastructure for low-density populations because its rollout may cost 80% higher than in urban zones [17].

So far, several approaches have arisen to reach the desired coverage index. Knowing that 5G services are not considered for the countryside yet, IoT services are limited to highly reliable networks. More quantity of unfolded BSs and more coverage index may be reached, increasing efficiency [16]. The BS coverage area is greater than 0.5 Km, and having enough overlap with adjacent cells will ensure the quality of roaming at the maximum allowable distance among them [18].

In mobile networks, the handover parameter is triggered when a user equipment (UE) detects a better signal strength of the neighboring cells [1], but it can also be regarded as non-convergent in the case of rural zones. Thus, identifying the BS coverage area at network planning is a relevant part of the design process. 3G services may be the first technology to be implemented in the countryside since it is possible to monitor the network parameters—such as coverage and cell capacity—by implementing appropriate Signal-to-Noise rates (SNR) and QoS index. It is important to recall that the rural connectivity gap is proportionally greater for low-income households [19].

After reviewing some references, we found that *Stratospheric Communication Platforms* (SPC) have been trending in the last decade for outdoor solutions [20,21]. The Loon Project searched for possibilities in building a new layer for the connectivity ecosystem in the stratosphere based on weather balloons with distributed self-optimization [22]. The Loon LLC group tackled the challenge of extending internet access worldwide based on this approach until the project was closed down in 2021 [23,24]. Another intended sample was Facebook Aquila; however, it collapsed in 2018 [25].

Another kind of alternative to cover rural populations includes the use of LEO satellites. Moreover, LEO and SCPs significantly enable coverage increase and do not require new terrestrial towers. Therefore, these options can offer highly reliable data rate services while demanding simple but special maintenance attention with respect to its tracing [26]. Figure 3 states a feasible implementation of the reviewed solutions in the countryside for outdoor areas, aiming to develop new tendencies considered in Section 2.3.

Figure 3 entails the use of alternative-to-terrestrial wireless connectivity aimed at rural coverage. To promote the solutions mentioned previously, the SCPs bring several attributes in offering a low-cost and widespread array of services in the countryside. Among them, in [27], the author listed the advantages of deploying these airships—either a constellation or a singular aerial unit—which may be summarized into three categories: seamless countrywide coverage, power consumption trade-offs, and a higher speed of transmission.

Since SCPs enable an extension of coverage, the unobstructed free-of-shadowing LOS roaming service would boost roaming services [27], leveraging the autonomous and discrete features of these devices. For instance, Espinoza et all. led a feasibility study in [28] for two Peruvian rural areas to promote connectivity, aiming at the underserved/unserved communities despite the rough landscape of either the Andes range or the Amazon rainforest. The authors used stratospheric balloons to serve the growing customer and speed demand for up to eight years. After that, an LTE-based network complemented the established structure.

Nonetheless, the fully fledged satellite-based networks carried on a cooperative relay scheme to accomplish the ongoing high demands of reliable, seamless, and high-rate transmissions. Therefore, the space-air-ground integrated network [29]—named SAGIN—combines LEO satellites, SCPs, and terrestrial BS to achieve the future requirements of 6G communications by addressing the provided-by-satellite coverage probabilities enhanced with terrestrial gateways as the authors in acknowledged in [30] in addition to real-time dynamic adjustment of HAP-based networks.

### 2.2. Indoor Case

Indoor-improving techniques outlined the strategies that enhance user experiences inside closed spaces. Therefore, there is more interference resulting from physical obstacles. This case requires evaluating the best estimation of indoor coverage provided, looking for optimal system planning. The feasibility in implementing algorithmic solutions based on UE location estimation appears to be challenging since their location accuracies depend on the integrated sensors in devices used by authors in [31].

Satellite-based networks and other high-altitude platforms suffer excess losses because the slant path intersects several obstructions than compared to terrestrials. Nevertheless, using repeaters at lower frequency bands—despite the bandwidth limitation—can fulfill the requirements demanded from users [10]. These devices are low-cost and readily available, hence boosting signal propagation and simultaneously enhancing indoor coverage. Figure 4 shows a potential indoor-improvement deployment for a satellite-based backhaul.

In addition to satellite-based backhaul solutions, other approaches outperform the repeater’s throughput, particularly for IoT devices and MTC at a glimpse, and Section 2.3 foresees the need reinforce this limitation. On one hand, the authors in [32,33] analyzed the development of indoor coverage—focused in suburban and rural areas—by deploying LTE-M, NB-IoT, LoRa, and GPRS solutions to cover a greater zone. On the other, there is a reduction in indoor time dispersion by dynamical estimation of the channel parameters in full swing machine learning techniques [34].

Regarding the first approach, Lauridsen et. al contrasted the technologies in a particular Northern Denmark region area by assessing their supported Maximum Coupling Loss [32,33]. The results accomplished 99% of indoor devices coverage for both LTE-M and NB-IoT solutions as long as these devices experienced 10 dB additional loss at its maximum level.

In the case of deep indoor devices—namely basement or underground located—NB-IoT performed the best, with 8% of outage probability, followed by LoRa (13%) and GPRS (60%) considering 30 dB additional losses. Despite the outstanding outcomes to shed light on optimal indoor coverage, the authors suggest a throughput evaluation of pico-cells or Wi-Fi access points to enhance connectivity inside closed spaces [32].

### 2.3. New Services

A few years ago, trending services such as IoT and 5G were considered challenging to implement in rural areas, especially for Latin America because there were no considerations to grant a reliable and high-traffic supported backhaul network. Nevertheless, these paradigms would hook up dispersed nodes located in remote zones nowadays, with staggering downlink/uplink rates, aiming to accomplish the requirements for MTC and Narrowband Internet of Things (NB-IoT) [35].

IoT promises to be a suitable technology for upgrading the countryside—a stable network may be guaranteed—following the massive number of connected things and the heterogeneous nature of IoT devices. On the other hand, there is the incursion of MTC application domains, such as agriculture management, transportation, logistics improvement, and crop automation, being one of the fastest-growing telecommunications technologies, especially in urban contexts [36]. LTE-based MTC addresses advantages in increasing capacity, traffic response, and spectral efficiency [37].

Diverse strategies have arisen from assessing the most appropriate technologies to furnish high-speed broadband and to reach desired standards such as service speed and setting up at 30 Mbps in European rural areas. Ioannou et al. in [38] stated that FTTdp (Distributed Point) solution using the G.fast standard is a cost-effective alternative to VDSL, which is the current widespread technology in Europe granting connectivity in the countryside. The authors acknowledge that FTTdP G.fast readily enables bandwidth upgrade, but the model is not cost-efficient in terms of investing in geographically sparse populations [16,17].

Consequently, LTE Fixed Wireless Access Networks (LTE FWA) could be an available, attainable solution, bearing in mind extensive LTE infrastructure in a significant rural part of the world. Regardless of whether newly emerged 5G standards are desirable for implementation, we can upgrade LTE FWA through the LTE-NR model, which creates a tight interaction between LTE and the new radio system. The also known model of E-UTRA-NR Dual connectivity—or EN-DC—allows benefits in aspects of user throughput in both low and high traffic load conditions [39].

Foreseeing the inclusion of the services mentioned above, the design of internet access solutions should be engaged with the three main factors as the authors outlined in [40]:Affordability—for avoiding undue hardships by employing reliable networks;Social shareability—to gain access through selfless (shared) connections;Geographical network coverage—where networks allow the user’s mobility by themselves.

Simultaneously, the requirements on ubiquitous coverage will not follow the one-size-fits-all standard to pursue a more connected rural society [41]. Figure 5 summarizes the information granted by the GSMA’s reports [42,43], which attempt to state the main driven innovations through an improved roll out in three foremost aspects:BS infrastructure—far-flung from the traditional macrocells model;Backhaul planning—avoiding the higher cost of urban deployment;Energy—mixing up with renewable sources;Blue Sky solutions—although those remain at the proof-of-concept stage.

These innovations will move beyond the traditional business model—such as *CapEx*—and local governments should create new regulation principles to harvest investment in network infrastructure.

## 3. Potential Solutions

There are several challenges to face in rural areas in terms of reliable and enhanced mobile networks. This need triggers the state-of-art study of diverse network models for the countryside in order to introduce ubiquitous solutions in which connectivity is available at anytime and at any location with respect to the population’s demands in a fully connected society.

By the first attempts to overcome likely hardships, such as insufficient population for deploying infrastructure, adaptive solutions struggle with the current unfolding of Mobile Network Operators (MNO). The new platforms or devices—that enhance coverage and other rural Key Performance Indicators (KPIs)—leverage practical alternatives for outdoor environments.

There have been studies that cater to rural coverage by using the TVWS-spectrum sharing approach that utilizes free UHF band channels from analog switch-offs at a specific time and space location [15]. Indeed, the primary user (PU) exclusively uses frequency resources on bands 470 MHz and 710 MHz.

On the other hand, S. Hasan et al. [44] aimed to recover GSM whitespace—or the non-actively used and licensed GSM spectrum—to support dynamic spectrum sharing, hence achieving a suitable QoS would not be attached to low throughput and high latency. Regardless, other kinds of solutions have arisen so far that aim for a fully connected countryside.

In the following subsections, several trustworthy approaches will be set forth for diverse rural outdoor solutions, such as unmanned aerial vehicles (Section 3.1), low altitude platforms, and high altitude platforms, and satellites (Section 3.2). Then, Figure 3 graphically summarizes the solutions as mentioned above to cope with rural not-spots.

### 3.1. UAV-Assisted Networks

Nowadays, unmanned aircraft have commercial uses and have enabled new research interest and innovation toward improving connectivity. The smaller the airship, the better the performance in bestowing coverage, especially for isolated areas. In this case, the drone industry has addressed several civil instances and applications within an affordable and straightforward aim: leveraging UAVs’ maneuverability to readily provide connectivity as an off-the-shelf alternative within the current MNO infrastructures.

Historically, the first purpose for Unmanned Aerial System (UAS) was for military and surveillance fields. During the second half of the 20th century, Warfighter’s internet yielded a reliable and readily deployable UAV-based ad hoc network to boost backbone communications [45]. This exploited UAS approach resulted in higher throughput standards. Therefore, a network-centric UAS operation concept arose beyond military and political boundaries and was consequently adopted for civil and economic interests. In a nutshell, the unmanned airships outpaced soldiery endurance.

Since the use of drones has been expedited, the need for regulating them has arisen as well in terms of complying with safety standards, even though they reach lower altitudes than other larger forms of aircraft. Therefore, the Global Unmanned aircraft system Traffic Management Association—or GUTMA—appears to foster trustworthy, secure, and efficient integration of UAS into global airspace, addressing drone stakeholders practices—defined as UTM stakeholders—by close cooperation and continuous flight information management [46].

To foster a collaborative and innovative community for UTM stakeholders, GUTMA encourages governments to adopt operation-centric, safe, fair, and secure deployment of UTM solutions. Moreover, with respect to allowing the full integration of UTM services with the current network infrastructure, the first step should predict the digitalization needs of UAS technology trends [47]. Once these are set forth, Table 2 compiles some of the key specifications for the UAV-assisted network in line with the deployment scenario, namely in urban, suburban, and rural contexts.

The gathered information in Table 2 divides the network features into two correlated fields: target scenarios and use case context. Concerning the first field, Muruganathan et al. approached the stakeholder populations in [48] and their LTE network’s technical deployment in environments such as urban-macro with aerial vehicles (UMa-AV), urban-micro with aerial vehicles (UMi-AV), and rural-macro with aerial vehicles (RMa-AV). The last scenario exhibits a better mobility performance (BMP) than the others. The second field considers zone density, emphasizing the highest (HD), the medium-to-high (M2H), the low-to-medium (L2M), and the lowest (LD) densities [49].

An analysis of coverage issues should extend the operational scope through defined network architecture to successfully deploy aerial communications. A likely first option launches the UAV model by hooking up one or several Ground-BS (GBS) and using the drone as a relay node in the network. Secondly, a swarm of drones seems suitable for covering a vast extension of nodes or rural-dispersed nodes, creating a solid construction of flying ad hoc network (FANET) networking. The last strategy outpaces the challenging issues that mobile ad hoc Networks (MANET) tackled in terms of communication range since a ground node can indirectly communicate with other hops through several aerial relay nodes such as UAVs [50].

The concept of FANETs has arisen in the literature in order to top off a particular form of Vehicular Ad Hoc Network (VANET) communications and addressing scalable, reliable, real-time peer-to-peer mobile ad hoc networking between aerial and ground nodes [51]. Table 3 relates some UAV-based communication surveys where the authors have thoroughly reviewed UAS modeling strategy in fields such as civil, security, and traffic management, among others.

The approaches, as mentioned earlier and among others, are comprehensively explained in Table 4 and Table 5. The first acknowledges the literature of UAV-based networks between twenty and five years ago, which states the strategies that the cited authors assessed for expanding MANET coverage primarily by algorithmic solutions. The second leads our survey to the outstanding aim: to gauge the promising models for rural communications, raising the current cellular infrastructure, or even adopting a new topology for ubiquitous coverage.

#### 3.1.1. The A2G Channel Modeling

W. Khawaja et al. comprehensively addressed the available A2G channel models—also including the timeliness to be extended even for Air-to-Air (A2A)—in the study given by [56]. Throughout the paper, they stated the most relevant UAV’s A2G schemes and thereby identified the limitations and future research directions for UAS-based communications scenarios. An outstanding classification of the analyzed literature includes deterministic methods, stochastic modeling, and their fusion.

Even though the performed analysis provided by Khawaja et al. remarkably spurred the development of a foremost deep-study on the A2G channel technical model, it is relevant to include a third slope into the classification: Machine Learning (ML) approach, which is scarcely investigated in the article [56]. To shed light on this end, Table 6 attempts to state some leading research studies to sketch out the aerial channel beneath two main strategies, ML and Geometry-based (GM) scheme, in addition to setting forth whether these strategies lie in stochastic (St) or non-stochastic (N-St) approaches.

#### 3.1.2. Regulation

The 3GPP Association mainly tackles the protocols and regulations for UAS-FANET communication beneath the addressed need of the quickly maturing sector [103]. Consequently, in the eagerness to state new studies and new features for safe operations, there has been joint work with GUTMA, even involving the novel 5G framework use cases. To our best knowledge, Figure 6 introduces the areas that are being addressed in the latest 3GPP Releases, from Release 15 to Release 17.

There are other institutions concerned with developing UAS standards, such as GUTMA/GSMA, ASTM International, IEEE, ISO, EUROCAE, IETF, and JARUS [47]. For a handy insight on network safety, by avoiding a loss of service due to their proximity, we have briefly recapped the 3GPP-suggested edges [103] in the Listing 1, as long as new releases emerge in supporting the LTE aim [48].

**Listing 1 sensors-21-08037-t0L1:** 3GPP Releases Outline involved in UAS Communications.

Release 15 addressed the research studies about the ability for UAVs to be served using LTE networks in addition to a comprehensive analysis of potential interferences between eNodeB and UAS.Release 16 has an overview of the potential requirements and use cases to enable the necessary connectivity between UAS and UTM.Release 17 approaches the use cases and requirements for UAS identification and tracking beneath the application layer. It also gathers the 5G connectivity needs of drones in new KPIs into a 3GPP subscription.

### 3.2. Other Engaging Solutions

We have thoroughly reviewed the implications in assisting rural networks by employing UAVs; moreover, other engaging solutions can enable broad coverage in the countryside and in shedding light on its connectivity. On the onset of the first decade of 2000s, SCPs appeared to be a prominent answer for fixed and mobile applications. These devices remarkably outpaced the unprofitable gap since they have arisen as a cost-effective solution for urban, suburban, and rural areas [21].

Aside from dedicated area coverage independence, the authors in [21] pointed out that Sky Station platforms may provide higher capacity—by higher frequency reuse—-than other wireless systems, the possibility of grant enhanced roaming, and the possibility of choosing their stationary point. Another seamless option for rural connectivity includes satellites, namely LEO configuration. The following subsections will deepen the strategies mentioned earlier, and UAVs also fall into this category.

#### 3.2.1. Altitude Platforms

The altitude platforms are grouped into LAPs (Low-altitude platforms) and HAPs (High-altitude platforms). Song et al. in [104] granted the main difference about the prior categories. LAPs gather the aerial platforms at an altitude of 20 km. UAVs, drones, and blimps fall into this group since they cannot support higher payload capacities, and their autonomy relies on SWAP constraints [105]. As Section 3.1 discussed, UAVs can perform far-flung coverage, increase redundancy, and increase survivability, leveraging the swarm FANET architecture [104].

LAPs have lent dynamic and scalable networks that can quickly cover broader regions, although there are by-payloads stuck. In this case, there are two method to limit this situation: First, developing a suitable propagation model that includes the elevation angle—deployed at several altitudes—along with the MIMO output antenna diversity gain, especially for the last mile connectivity [106]. In the UAV case, the strategy may contain a formulation of statistical assessment of A2G propagation by either using Ray Launching or Ray Tracing geometrical optics models [107]. Second, Drone-to-Drone communication arises as a reliable collision avoidance system [105].

On the other hand, HAPs operate in a quasi-stationary position at an altitude of 20 to 50 km, becoming a viable option to furnish capacity and coverage enhancement [104]. The authors in [108] have envisioned these platforms as a super macro BS (HAPs-SMBS) to unfold high-traffic-volume networks in a metropolitan area in bargaining with smart city paradigms. Facing the LEO constellation shortcomings, HAPs-SMBS can mask the high path loss and high mobility effects.

The potential uses of HAPs—to tackle rural not-spots—shed light on dynamically managed radio resources and mitigate the crossed interference [109]. The rural environment has admitted more prevalence to network coverage instead of higher capacity density. The reason for this is that HAPS requires lower investment and provides high quality—even providing higher terrestrial QoS—and this alternative has been carried out to cover rural and remote areas [110]. At this point, the likely exploitation of radio environment maps and artificial intelligence in the ongoing infrastructure may allow a radius coverage area of more than 30 km, as Chukwuebuka highlighted in [110].

#### 3.2.2. Satellites

Satellite-based architecture has furnished an outstanding architecture to hook up the highly dispersed and remote rural nodes due to their scalability and flexibility in reaching vast geographical areas. In function of the developed network scope, the satellites’ orbit relies on defined classification [111]: LEO (altitude between 500 km and 2000 km), MEO (altitude into the range 5000 and 20,000 km), and GEO (altitude of 35,800 km).

Underneath the condition of service-as-primary-resource, LEO architecture, on the one hand, solves latency issues [112]; on the other hand, it has added remarkable bit rate capacities by multi-beam technology [113]. In contrast, e.g., GEO holds limited these parameters. Heading to the best alternative for rural not-spots, LEO has become the best complementary structure of terrestrial networks in the countryside, figuring out several shouldered challenges, such as routing problems and raining attenuation [114].

In order to provide seamless and continuous service by LEO satellite networks, these approaches have adopted constellation shapes whereas QoS is guaranteed, fueled by novel routing protocols regarding UE location and exploiting deterministic LEO topology. Therefore, the route bottlenecks should be foreseen in any pair of end-users, as the authors said in [115]. By avoiding design planning deficiencies, the system’s user capacity becomes greater, and the covered geographical zone becomes larger [114].

## 4. Discussion

At this height, the rural zones have struggled to embrace fully fledged connectivity. Regarding Latin America’s situation, three considerable constraints are in conflict with the ubiquitous rural coverage aim: First, MNOs do not furnish a suitable telecom infrastructure outside of urban environments. Secondly, rural settlements are concentrated but geographically sparse, occupying common hot spots. Finally, the studied strategies should be based on bespoke hardware requirements since uneven relief and ecosystem variation hamper the static estimation of channel parameters—the latter demands higher investment cost—which seems unprofitable for ISPs and MNOs.

The not-spots affect directly rural inhabitants, especially those who attempt to foster rural businesses, which include mainly agricultural and new industrial activities in the countryside. Hence, Table 7 states the advantages and shortcomings of the studies solutions—in Section 3—while we spur ongoing research of UAV-assisted networks deployment driven by mobility, cost-effective, and the other leverages outlined in Table 7 that can bring networks to the uncovered regions. Further works include the analysis of dynamical propagation model and simulations of LTE—aiming for 5G-NR deployment—at incoming experimental stages.

### Future Research Opportunities

In the prior section, we have introduced three achievable solutions to strive against the countryside’s not-spots. There remain shortcomings stuck in the fully fledged method of granting connectivity to pursue endurance in the deployed system. UAV-based networks seem to be an attractive option due to their commercial affordability, as we pointed out in Section 3.1. However, for now, both UAS and Altitude Platforms have factors opposed to large scale use, such as payload capability—to shoulder the network equipment—and insufficient MNO interest, behind higher returning investment rates.

Consequently, three categories claim further analysis in the case of deploying aerial networks. Firstly, A2G channel modeling needs to be supplied by a realistic propagation model, since most of them are still limited to a single device or particular environments. Moreover, there is a considerable need to characterize the by-mobility Doppler effect, in addition to the channel’s captured time variation addressing more precision and accuracy. Fueled by the efforts to sketch the A2G channel—as it was analyzed in Section 4—the need to address research in this field has arisen and becomes more decisive as consumer demands grow.

Secondly, aerial platforms lacking an optimal 3D placement are of concern. A matchless location bestows the coexistence with the terrestrial cellular networks and avoids mutual interference with GBS. In the case of UAS, an optimal arrangement of UAV-BS can yield a minimum downlink transmission latency, setting up previously the drone-BS location and transmission bandwidth [52]. This approach can reduce the total flight time while also enhancing energy consumption.

With respect to concern on energy issues of SCP—which may be the foremost challenge to assure connectivity’s significant periods—the utilization of peer-to-peer energy sharing has been investigated since energy is a limited resource in mobile networks because they are jammed in non-renewable sources such as batteries. According to the application, an attractive solution to outpace the excessive consumption of energy, mainly focused on renewable sources, appears to be a significant research field for guaranteeing communication availability [116].

Thirdly, outstanding cellular network planning foresees the minimum number of required aerial nodes to cover a given geographical area, either partially connected or entirely disconnected. Hence, to maximize the total covered zones, there should be a previous identification of users and obstacles. Beneath this regard, prior frequency planning and signaling overhead analysis can assure greater network throughput, especially for high-frequency bands.

Finally, an in-depth design of the bespoke-solution construction affords countless advantages to aim for in a fully connected countryside. In this case, embracing an expected radius of 30 km [109]—or even a greater value—assures the coexistence of either LAPs or HAPs with terrestrial systems, and sharing the same spectrum can extend the coverage in rural and remote areas. Other strategies involve a novel antenna array design and aerial swarms or constellations, which are expected to be further researched in broader 5G investigations.

## 5. Conclusions

Nowadays, complete rural internet access may be an incongruous reality due to the lack of efforts to deploy a suitable mobile networks infrastructure. However, data demand has grown recently since many rural inhabitants consider using technology to improve their quality of life by implementing trending technologies, such as IoT. Although Latin American countries have recently envisaged closing the connectivity gap, there are remote geographical zones where not-spots are a significant challenge to governments because they strive to outpace inequality under the insight of fully fledged coverage.

Bearing in mind our study cases, in Mexico and Colombia, which have economically and technologically developed in the last decade, the connectivity gaps are noticeable. Therefore, implementing alternative and efficient solutions—as listed in Section 3—will involve hooking the peripheral population up with reliable deployments. The COVID-19 pandemic has accelerated the reshaping of a noteworthy need for connectivity since most of our performed activities leverage digitalization growth to partake in affordability and access. Although several rural populations remain fully offline, the recent efforts to stimulate new steady links have triggered new opportunities to access online education, employment, or critical health and sanitation advice.

We have summarized some strategies to strengthen connectivity in rural environments, especially for Latin American countries. By establishing statistics that best draw the mentioned phenomena, we encourage further access to ICT and encourage the target of providing affordable access to the internet in developed countries, which in turn considers rural and geographically remote populations. Hence, solutions such as UAVs, HAPs/ LAPs, and LEO satellites have arisen for most cost-effective bargaining. However, we have comprehensively studied UAS scope in communication because its efficient maneuverability can solve the coverage problem through a solid construction of either GBS or FANET approaches.

## Figures and Tables

**Figure 1 sensors-21-08037-f001:**
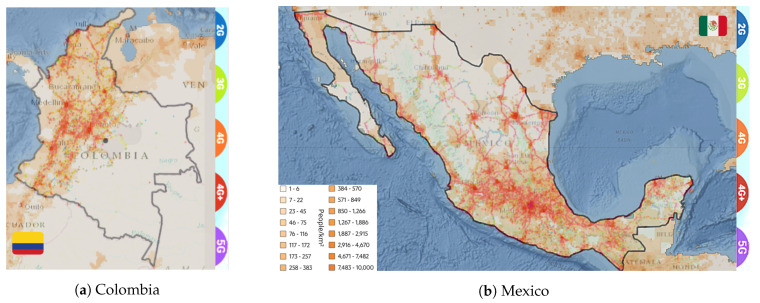
Correlation between population and current mobile network coverage in both study cases.

**Figure 2 sensors-21-08037-f002:**
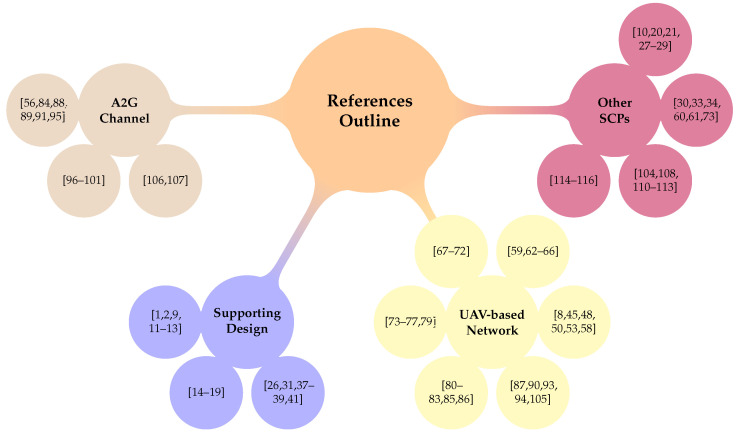
Some Used references in our survey.

**Figure 3 sensors-21-08037-f003:**
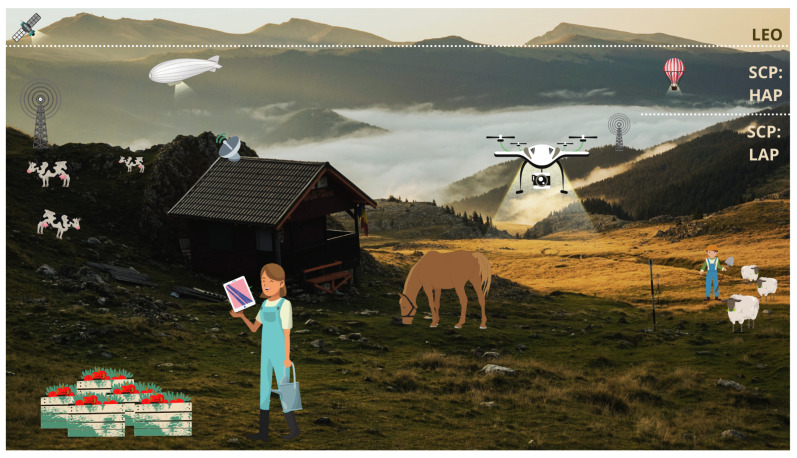
Some solutions for outdoor networks issues.

**Figure 4 sensors-21-08037-f004:**
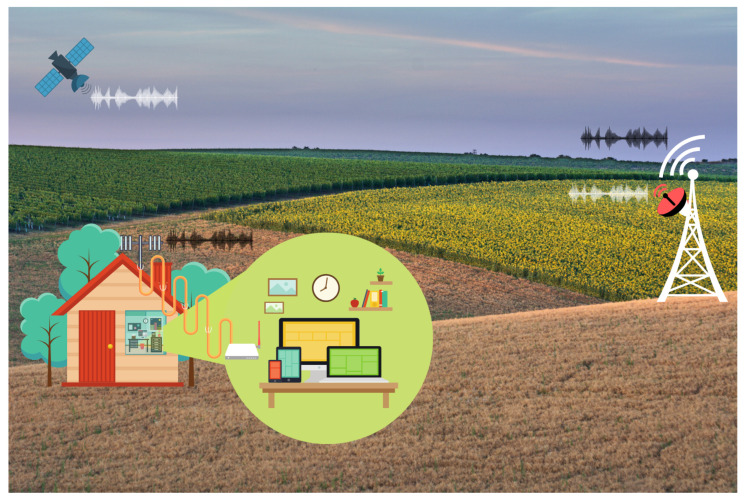
Indoor solution for a satellite-based network.

**Figure 5 sensors-21-08037-f005:**
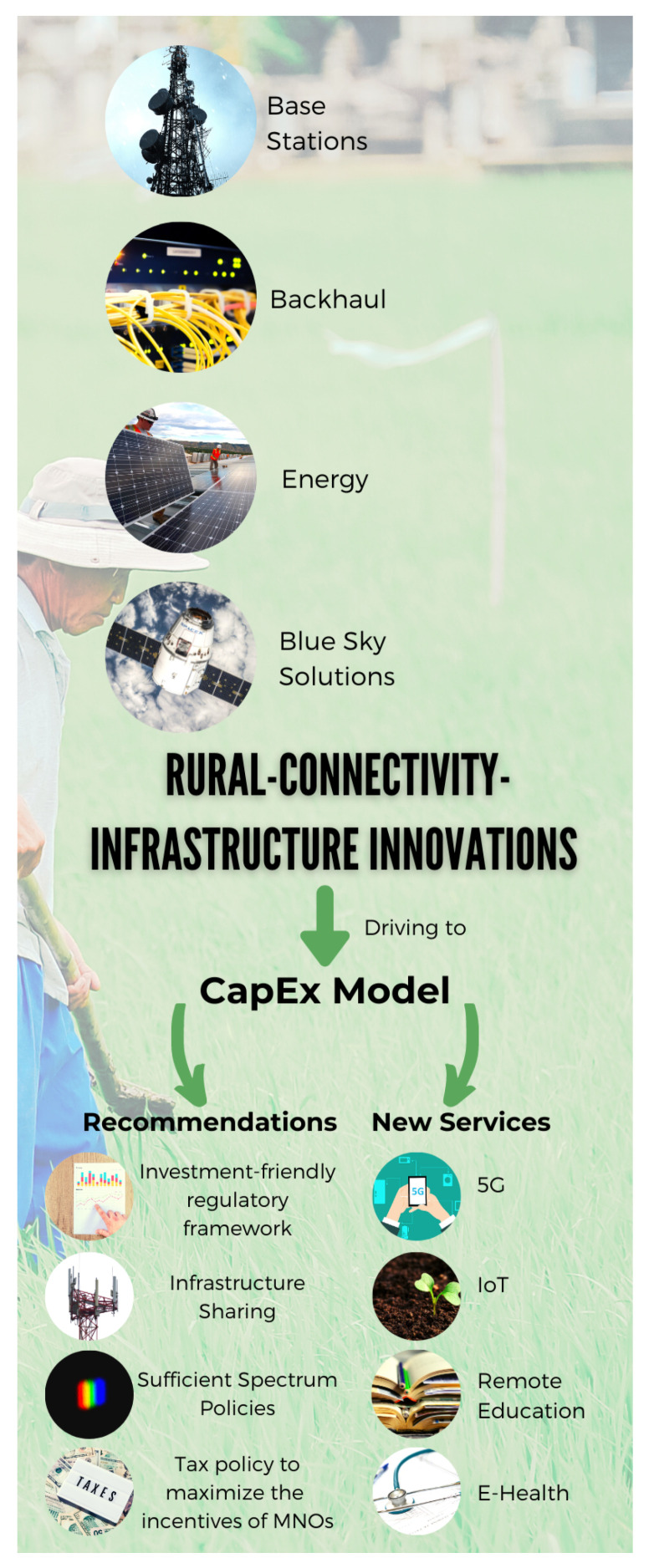
Innovations for rural connectivity.

**Figure 6 sensors-21-08037-f006:**
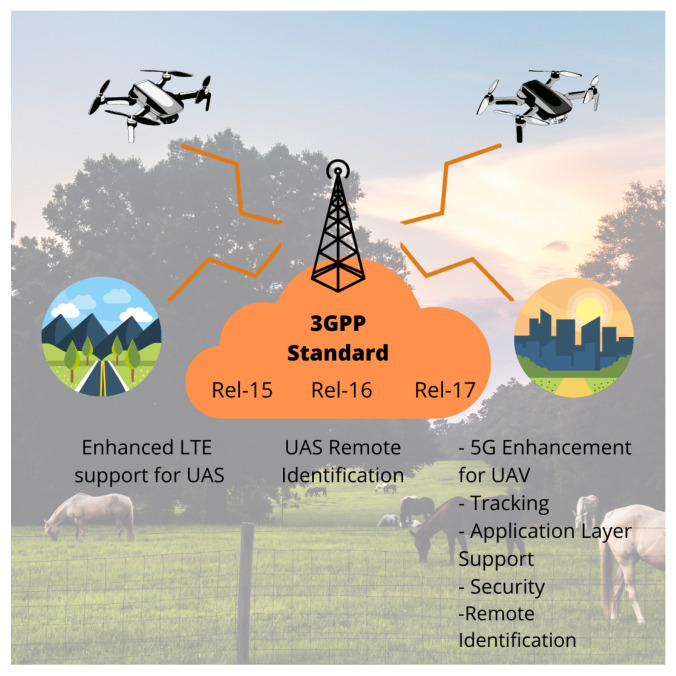
UAS addressing in 3GPP standards.

**Table 1 sensors-21-08037-t001:** Contrast between MCI and RPD of 15 Latin America countries [7].

Country	MCI	RPD
Argentina	67.2	8
Bahamas	68.7	17
Brazil	63.5	13
Chile	73.2	12
Colombia	63.7	19
Costa Rica	63.3	20
Dominican Republic	59.8	18
Ecuador	65.3	36
El Salvador	55.4	27
Haiti	32.8	44
Mexico	67.6	20
Panama	65.3	32
Peru	66.6	22
Uruguay	76.7	5
Venezuela	57.4	12

**Table 2 sensors-21-08037-t002:** Context-based Specifications for UAV Networks.

Scenario	Network Parameters					Context		
	LHT (m)	UHT (m)	BMP	LOS	NLOS	Use Case	Network Configuration	Flight Time (min)
UMa-AV	22.5	100		X		HD/M2H	5G	TBD
UMi-AV	TBD	TBD			X	M2H		15–45
RMa-AV	10	40	X	X		L2M/LD	LTE/LTE+	60–180

**Table 3 sensors-21-08037-t003:** Some surveys of UAV-based communications.

Publication	Brief Summary	Approaches Fields
Mozaffari et al. [52]	A group of potential benefits and applications of UAV-based communications in enhancing coverage, capacity, and reliability of wireless networks.	The key UAV challenges include 3D deployment, performance analysis, channel modeling, and energy efficiency. A comprehensive overview of potential applications, chief research directions, and challenging open problems, among others.
Li et al. [53]	A noteworthy integration of 5G technologies with UAV communications networks upon an emerging space-air-ground integrated network architecture.	Space-air-ground integrated network envisions for beyond-5G communications. 5G techniques for physical and network layer of UAV scheme and joint communication, computing, and caching.
Fotouhi et al. [54]	A development summary promotes the smooth integration between UAVs and cellular networks without a one-size-fits-all but affordable model.	The authors surveyed interference issues and potential solutions on UVA-based flying relays and BS approaches. The article sets forth the new regulations and protocols to grant cyber-physical security in both aerial nodes and UEs.
Shakhatreh et al. [55]	An exhibition of the next large revolution in civil applications by introducing UAV technologies to state feasible research trends and future insights.	Addressed civil applications: road traffic’s real-time monitoring, wireless coverage, remote sensing, search and rescue, surveillance, and civil infrastructure, among others. Discussed key challenges: charging, collision avoidance, security, and networking.
Khawaja et al. [56]	Modeling Air-to-Ground (A2G) propagation channels in designing and evaluating stages of UAV communication and links attempts to improve AG channel measurement campaigns.	AG wireless propagation channel research includes payload communications and control and non-payload (CNPC) networks. The AG channel study tackles limitations such as large and small scale fading.
Hayat et al. [57]	Aerial network missions should vary according to the civil application aims.	Search and rescue coverage Network coverage Delivery and transportation Construction

**Table 4 sensors-21-08037-t004:** Phases of UAV-based network models.

Phase	Approaches	Strategies Models	Advantages/Findings
**Early:** <**2011**	**Military** **Services**	Airborne Communication Nodes to form a backbone network for Warfighter’s internet [45].	▪ Allowing connection for separated forces ▪ Reliable and easily deployed
		The biologically inspired metaphor algorithm of bird flocking for UAV nodes’ placement and motion, adapting their mobility [58].	▪ Especially useful for rugged and mountainous terrains with heavy signal attenuation. ▪ Achieving a stable connection and load balancing.
		Dynamically placing UAVs considered as relays nodes to provide full connectivity in a disconnected ground MANET through heuristic and algorithmic approaches [59].	▪ Location tracking that allows an optimal interaction between ground nodes and UAVs without introducing new MANET protocols. ▪ Cost reduction based on finding the minimum number of needed UAVs.
	**Integrated** **Architecture**	Two-level Satellite empowered architecture (HAPs/UAVs + Satellite) to improve limited coverage, guaranteeing superior bandwidth access [60,61].	▪ Allowing interconnection with remote locations. ▪ Enhancing hot-spot coverage with low latency rates. ▪ Mitigation of shadowing impairments through a HAP/UAV repeaters configuration.
		Implementation of UAV-HALE (UAV-High Altitude Long Endurance) platform as a base station with an adaptive antenna array [62].	▪ Covering rural low-densely populated areas and isolated-by-relief regions. ▪ Support the telecommunication system in emergencies. ▪ Assist hot-spots traffic with a lower cost solution ▪ Provide higher QoS, increasing capacity, and keeping lower computational complexity.
		An algorithmic solution to state and hedonic coalition formation consisting of a determined number of UAVs continuously collecting packets from task arrays [63].	▪ Performance improvement based on the self-organization of air nodes and tasks into independent coalitions. ▪ UAVs can assess the decision to act as collectors or relays (to enhance wireless transmission). ▪ Suitable model to tackle several aims as surveillance or wireless monitoring.
		Evaluation of A2G links coverage using UAVs at altitudes up to 500 m performing as a radio relay platform in low RF environments [64,65].	▪ Support over 90% coverage of the ground receivers within 10 dB of LOS Path Loss. ▪ Excellent connectivity for low flying UAV in limited urban areas considering SWAP, even for building-blocked receivers. ▪ For higher altitudes, the coverage becomes homogeneous in rural zones.
	**MANETs** **Upgrade**	UAV-assisted MANET model, which is rooted in four connectivity regards: global message (successful propagation to all nodes), worst-case (dividing up a close network), bisection (division cost), and k-connectivity (failed nodes threshold before a disconnection) [66,67].	▪ The aerial nodes can generate, receive, and forward data packets or improve network connectivity and availability. ▪ The model will achieve better QoS and coverage. ▪ As the proposed method, an adaptive heuristic algorithm can provide a simple solution and reach better performances.
		Performance assessment of ad hoc routing protocols, such as GPRS, OLSR, and AODV, in the context of swarms of UAVs, also considering the relative location of destination nodes [68].	▪ Maximize the throughput with a minimum number of neighbors into the swarms to ensure connectivity. ▪ Minimize power consumption and optimize loiter time to prevent cross interference and redundant transmissions through spatial multiplexing technique.
		Ad hoc UAS-Ground Network (AUGNet) solution, where an Unmanned Aircraft provides additional connectivity for ground nodes driving into shorter routes with better throughput [69].	▪ Improve connectivity at the network coverage boundary. ▪ Introduce the net-centric UAS operation concept, a tight coupling between communications, mobility, and task fulfillment.
**Medium:** **2011–2016**		Mobility strategy for UAV-compound MANET to support communication data flow between ground nodes in a dynamic topology network [70].	▪ Provide the most appropriate air nodes position that maximizes network performance. ▪ UAV nodes can flexibly communicate with ground nodes in the LOS, covering a greatly extended area. ▪ Ground nodes periodically grant their communication status to the air-backbone to find the best mobile strategy.
		Analysis of the coverage problem to address several issues in UAV-FANETs, expecting to extend their operational scope and range and a reliable response time [50,51,71].	▪ The solid construction of FANET networking standards will result in scalable, reliable real-time peer-to-peer, new-form MANETs. ▪ Aimed at robustness of the coverage algorithms, considering the several constraints in these kinds of networks, especially for UAVs fleets. ▪ Cooperating UAV form aims to increase reliability for aerial missions, ensuring the connectivity of non-LOS systems.
	**Connectivity/** **Coverage** **Enhancement**	Neural-based cost function approach to improve coverage and boost capacity into geographical areas subject to high traffic demands [72].	▪ Provide reliable multi-connectivity using UAS overview as relays between a disconnected network and enhance connectivity. ▪ The model can provide better capacity, reliability, and prolonged connectivity to tackle the inefficiency in handling macrocellular network traffic demands.
		The connectivity-based mobility model (CBMM) compares coverage and connectivity performance, looking for an optimal tracing and sense of a given area [73].	▪ Monitor inaccessible or dangerous areas to deliver information with lack-of-infrastructure regions. ▪ CBMM allows adapting air node direction to maintain steady links to ground stations or their neighbors. ▪ Reduce the overlap between covered areas by using an efficient and limited number of UAVs with a specific spatial density.
		Efficient 3D deployment of multiple UAVs as portable base stations, seeking downlink coverage performance’s maximization in using a minimum transmit power and directional antennas [74].	▪ Aerial Base Stations have a higher chance of LOS links to ground users. ▪ UAVs can readily move and have a flexible deployment to provide rapid, on-demand communications. ▪ Using directional antennas, the model may enhance UAV-based networks because of effective beamforming schemes.
	**Deployment** **Focusing**	Low Altitude Small UAVs (SUAV) pilot provides a micro-scale mobile communication relay, attempting at a superior propagation model and increasing bandwidth reuse for emerging traffic hotspots [75].	▪ The model achieves an improvement of mean throughput (>22%) and QoS (>70%) in both rural and urban environments. ▪ Offer new possibilities for addressing local traffic imbalances and providing great local coverage.
		Deployment of Drone Small Cells (DSCs) or aerial wireless base station to optimize the covered area. In the presence of D2D users, new challenges—such as coverage performance—should be tackled [76,77].	▪ The optimal UAVs’ altitude results in maximum coverage and system sum rate simultaneously when introduced into underlaid D2D communications links. ▪ In the case of two or more DSCs, an optimal separation distance will grant maximum coverage for a given target area.
	**Civil** **Applications**	QoS requirements ranking of UAV networks marked into a practical choice for commercial applications. These aims will outline the design of emerging aerial networks [57].	▪ Delimitation of the missions into four categories: Search and Rescue, Coverage Expanding, Delivery/Transport, Construction. ▪ SUAVs have turned into handy but inexpensive options for commercial aims due to their their ease of deployment, low maintenance costs, high-maneuverability, and ability to hover. ▪ Wi-Fi technology can support several prior categories whether each application requires a few number of hops amongst the nodes.
		UAV-aided Wireless Communication may be a promising solution for scenarios without coverage infrastructure [78].	▪ UAV systems are more cost-effective than other solutions—such as HAPs and satellites—providing performance enhancement and adaptive communications. ▪ UAV-based networks involve three typical use cases: *ubiquitous coverage*, *relaying*, and *information dissemination and data collection*.

**Table 5 sensors-21-08037-t005:** Phases of UAV-based network models (continuation).

Phase	Approaches	Strategies/Models	Advantages/Findings
**Novel:** >**2016**	**Rural** **Panorama** **Addressing**	Energy consumption optimization aims to improve aerial node missions and connectivity in the countryside by using a graph-based structure [8], besides an optimized model called RURALPLAN [79,80].	▪ The multi-period graph approach derives into Genetic Algorithms. It guarantees the coverage and efficient management of UAV consumed energy. ▪ RURALPLAN can reduce energy consumption by up to 60%. ▪ The deployment of UAV-based networks can adopt a short-distance LOS, decreasing installation costs. ▪ By considering a set of optical fiber links to support the backhaul network, the capital and operation expenditures can be compensated, simplifying the stated model.
		Analysis of joined-architecture networks, mixing UAVs and GEO/LEO satellites, to increase radius coverage and state the usability of aerial nodes to assist fixed-infrastructure networks in the countryside [81,82].	▪ The use of aerial nodes, acting as relays, can cover vast rural extensions, addressing further mobile network generations—such as 5G—to implement steady-links IoT devices. ▪ Bearing in mind the optimizing cellular networks aim in the countryside, heritage functionalities of LTE can achieve prominent coverage radius in the sub-1 GHz bands, raising RF propagation. ▪ Since Non-Terrestrial Networks may be an integral part of 5G infrastructure, UAVs become the bedrock of a mixed-architecture network, especially in collecting data in massive MTC types of application.
		LTE networks can provide coverage by UAV nodes in rural areas, chiefly to boost the Command and Control downlink channel, despite the raised interference due to height dependency [83,84].	▪ The dependency of the large-scale path loss on the drone’s height may be challenging for achieving significant growth in coverage level, boosting the aerial-node’s perceived interference level. ▪ Applying network diversity, it is possible to improve the network coverage level and its reliability, since SINR would be better than the achieved −6 dB index under the full-load assumption. ▪ The interference conditions—because drastically changed UAV height— will determine channel characterization to assess wireless remote controls for the aerial nodes.
		Boosting aerial coverage of rural area network deployment to clear limitations by interference mitigation techniques [85].	▪ Interference canceling and antenna beam selection are strategies to improve overall—aerial and terrestrial— system performance. ▪ The abovementioned schemes will gain a 30% of throughput and achieve a 99% reliability increase. ▪ Downlink and Uplink radio interference trigger poor performance within aerial traffic.
		A Non-Orthogonal Multiple Access (NOMA) layout for UAV-assisted networks to provide emergency services in rural areas [86].	▪ The proposal carry out the performance of terrestrial users enhancement, resulting in a by-device that is consumed in energy minimizing. ▪ The proposed user-centric strategy follows stochastic geometry approaches for terrestrial users—placed into Voronoi cells—served by UAVs, achieving the location model of both nodes and UEs. ▪ In the case of the NOMA-assisted multi-UAV framework, the analysis of coverage probability can aim to properly set up the network’s power allocation factors and targeted rates.
	**Cellular** **Network** **Advance**	Optimization of the UAV-mounted base stations (MBSs) placement, setting forth a Geometric Disk Cover (GDC) algorithmic solution, which coats with all ground terminals (GTs) in an inward spiral manner [87].	▪ The correct deployment of MBSs can cover a set of k nodes with a minimum number of disks of a given circular surface with radius *r*. ▪ The computational complexity may be significantly reduced when coverage starts from the perimeter of the area boundary.
		The Path Loss (PL) Characterization for urban, suburban, and rural environments enhances the access technologies for low-altitude aerial networks, considering UAV height effects on the channel [88,89].	▪ By introducing a Correction Factor (CF), which relies on the UAV altitude, the large-scale fading and the PL of the A2G channel will be accurately characterized. ▪ In urban contexts, PL increases with horizontal distance. In the case of rural zones, PL is irrelevant to UAV heights, albeit it approximates to free-space propagation models at heights around 100 m. ▪ UAV-based networks face a large amount of neighboring interference due to the down-tilted antenna pattern of cellular networks. Moreover, the coverage behavior will be affected beneath this scheme.
		Improvement of coverage and capacity for future 5G configurations of aerial networks beneath two algorithmic approaches, entropy-based network formation [90] and latency-minimal 3D cell association scheme [91].	▪ By correctly selecting the UAV controller and then performing network bargaining, the aerial base station could top off a more remarkable improvement on its throughput, SINR per UE capacity in the order of 6.3% and minimal delays and error rates. ▪ With the increase in simultaneous requests within the next-generation heterogeneous wireless network, entropy approaches appear to be suitable for overcoming UAV allocation and Macro Base Station decision problems. ▪ Lifting 3D configuration for aerial cellular networks, a yield of reducing up to 46% in the average total latency would enhance spectral efficiency.
		Optimal design of aerial nodes trajectory in cellular-enabled UAV communication with Ground-BS (GBS) subject to quality-of-connectivity constraints about the link GBS-UAV [92].	▪ The optimization problem converges in a non-convex approach to find high-quality approximate trajectory solutions. ▪ Channel’s delay-sensitive rates and SNR requirements restrict the target communication performance. ▪ UAV’s mission completion time may guarantee an efficient method for checking the strategy’s feasibility.
		Cooperation of small and mini drones can further enhance the performance of the coverage area of FANETs—even other aerial-kind networks—by establishing a hierarchical structure of efficient collaboration of drones [93,94].	▪ In the case of ultra-dense networks, the approach efficiently broadens the common issues such as sparse and low-quality coverage and the non-steady aerial links. ▪ The rapidly unfolding of UAV carries out in the non-dependency of geographical constraints and implies system performance lifting by establishing LOS communication links in most scenarios. ▪ Among other advantages—at the top of cooperative distributed UAV networks— are the distributed gateway-selection algorithms used and stability-control regimes.

**Table 6 sensors-21-08037-t006:** Some efforts addressed in A2G modeling.

Cite	Approach		Scenario			Method		Aim	Contributions
	ML	GM	UMa	UMi	RMa	St	N-St		
[95]	**X**		**X**	**X**	**X**	**X**		PL and Delay Spread prediction for mmWave channels.	Low computational complexity. Full feature selection scheme. Frequency/scene-based transfer learning model.
[96]	**X**		**X**	**X**	**X**	**X**		PL and Shadowing effects analysis in 3D-LOS/NLOS Channel.	Unsupervised learning clustering technique to derive a 3D temporary channel.
[97]	**X**		**X**			**X**		PL empirical prediction with environmental parameters.	Location-based method by using 3D-GPS coordinates. Learning phase includes atmospheric conditions.
[98]	**X**		**X**	**X**	**X**		**X**	Collaborative algorithm to solve communication overload by achieving 1.5x throughput.	Optimization of Multi-UAV user deployment based in modified K- means distribution and POO.
[99]		**X**	**X**			**X**		3D non-stationary geometry-based stochastic channel model for A2G.	3D arbitrary trajectories. 3D antenna arrays for 5G. Computational Methods for time-variant channel parameters.
[100]		**X**	**X**			**X**		A MIMO wideband truncated ellipsoidal-shaped method with scatterer consideration.	Statistical derivation of space-time-correlation function and Dopler power spectrum density.
[101]		**X**		**X**		**X**		Geometrical model for UAV flight’s Multi-Path Components evolution.	Geometrical parametrization for the main MPCs. Simulation under non-intuitive effects of propagation.
[102]		**X**	**X**	**X**	**X**		**X**	Spatial-temporal correlation in function of UAV’s hover radius, flight altitude, and elevation angle.	Numerical approach of PL, Multi-shadow fading, Doppler shift, and channel correlation. Fixed-Wings UAV-BS Model.

**Table 7 sensors-21-08037-t007:** Comparison among the analyzed solutions for rural coverage.

Solution (Section)	Advantages	Disadvantages
UAVs [Section 3.1]	Easily deployable and portable. Reliable infrastructure to enhance coverage. New security standards by new routing protocols. Compatible with others as terrestrial and aerial network’s platforms.	Static-channel-modeling intermittent connectivity. Energy constraints and limited effective payload. Uncertainty on legislation. Inefficient obstacle awareness rollout.
HAPs [Section 3.2]	Commit to cover immensely inaccessible areas. Allows adaptable resource allocation. Low roll-out costs. Guarantee connectivity by a single platform. Agile deployment. Payload upgrading.	Few protocol standardization. Unfit design of traffic aggregation. Poor raters of interference mitigation in shared spectrum.
LEOs [Section 3.2]	Enable higher QoS than terrestrial. Reach a latency issue standard. Add significant bit rate capacity. Provide high capacity backhaul.	Insufficient coverage time assessment. Higher cost of deployment and maintenance. Most affected by fading effects. Unreliable communication at low elevation angles.

## Data Availability

This article has retrieved information from GSM Association Intelligence databases and reports. Both global MCI and RDP may be consulted through 2020-GSMA Mobile Connectivity Index. © GSMA Intelligence 2004-2021.

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
