# Peer review of "Closing Connectivity Gap: An Overview of Mobile Coverage Solutions for Not-Spots in Rural Zones"

_sensors, 2021, doi:10.3390/s21238037_

Round 1

Reviewer 1 Report

Figure 2 is confusing. Is the grouping of references to each of the categories into the clusters meaningful beyond easier to read?

Several places in the article, there are words that do not seem to fit with the message: "paradoxial", "unleashes" etc. This should be checked and corrected.

Author Response

Dear Reviewer,

In the below lines, I present the point-to-point response to each suggestion besides expressing my sincere gratitude for all the provided comments.

  • Point 1: Figure 2 is confusing. Is the grouping of references to each of the categories into the clusters meaningful beyond easier to read?

Response 1: Figure 2 introduces to lectors a straightforward split of the surveyed literature and thus addresses the paper's core beforehand. By considering the suggestion of turning the chart into more meaningful, the modification attempts to highlight the references that fuel the article's aim: stating the use of UAVs as an outstanding solution for not-spots.

  • Point 2: Several places in the article, there are words that do not seem to fit with the message: "paradoxial", "unleashes" etc. This should be checked and corrected.

Response 2: The use of unconventional words was restricted to allow the lectors an effortless understanding. Terms such as paradoxical turned into incongruous, and to unleash, into to rely on.

Reviewer 2 Report

This paper comprehensively investigates the use of three platforms to deliver broadband services, i.e., Unmanned Aerial Vehicles (UAV), Altitude Platforms (APS), and Low-Earth Orbit (LEO) satellites. It also indicates that UAVs are considered a noteworthy solution to aboard the rural zones for their efficient maneuverability and not-spots.

  1. Figure 1 shows the relationship between population density and mobile network coverage, but we cannot intuitively see the changes in population density. It is recommended to add a legend of population density changes to highlight this connection.
  2. The solution to rural communication coverage problems such as APS and LEO is the main part of author's explanation. Therefore, in Section 2.1 Outdoor Perspective, it’s suggested more literature survey on LEO and SPC.
  3. In Section 2.2. Indoor Case section, apart from the solution strategy of using repeaters at low frequencies, it’s suggested that some other indoor situation solution strategy research should be added as choices and comparisons.
  4. In section 3.1. UAV-Assisted Networks, the UAV-to-ground communication is analyzed. It is recommended to add some contents about the communication link model of U2G or air-to-ground. Actually, there are quite a lot papers about this topic, such as   1) Zhu, Y. Wang, K. Jiang, X. Chen, et al, 3D non-stationary geometry-based multi-input multi-output channel model for UAV-ground communication systems. IET Microwaves, Antennas & Propagation, 2019, 13(8):1104-1112.        2)Yang, Y. Zhang, Z. He, et al, Machine-learning-based prediction methods for path loss and delay spread in air-to-ground millimeter-wave channels, IET Microwaves, Antennas & Propagation, Vol. 13, no. 8, pp. 1113–1121, July 2019.
  5. Figure 1 is fuzzy and it should be modified.
  6. It is recommended to add the SPC and LEO methods to the outdoor scene to make Fig.3 more complete.

Author Response

Dear Reviewer,

In the below lines, I present the point-to-point response to each suggestion besides expressing my sincere gratitude for all the provided comments.

  • Point 1: Figure 1 shows the relationship between population density and mobile network coverage, but we cannot intuitively see the changes in population density. It is recommended to add a legend of population density changes to highlight this connection.

Response 1: Accordingly, Figure 1 was totally modified, in order to set forth a graphical correlation between population density and the covered zones in both study cases. The recommended legends were also included.

  • Point 2: The solution to rural communication coverage problems such as APS and LEO is the main part of author's explanation. Therefore, in Section 2.1 Outdoor Perspective, it’s suggested more literature survey on LEO and SPC.

Response 2: By coping with the lack of literature in Section 2.1, four additional references—Lauridsen, M et al, Aldossari, S et al, and Talgat, A—enhance the perspective of including SCPs and LEO satellites as feasible solutions for not-spots at the rural outdoor panorama.

  • Point 3: In Section 2.2. Indoor Case section, apart from the solution strategy of using repeaters at low frequencies, it’s suggested that some other indoor situation solution strategy research should be added as choices and comparisons.

Response 3: By undertaking comprehensive research on the indoor case—besides the use of low-frequency repeaters—LTE-M, NB-IoT, LoRa, and GPRS solutions were analyzed and compared for the most suitable technology to address the uncovered closed spaces. On the other hand, there is the timeliness of ML techniques implementation to achieve the identical end.

  • Point 4: In section 3.1. UAV-Assisted Networks, the UAV-to-ground communication is analyzed. It is recommended to add some contents about the communication link model of U2G or air-to-ground. Actually, there are quite a lot papers about this topic, such as 1) Zhu, Y. Wang, K. Jiang, X. Chen, et al, 3D non-stationary geometry-based multi-input multi-output channel model for UAV-ground communication systems. IET Microwaves, Antennas & Propagation, 2019, 13(8):1104-1112. 2)Yang, Y. Zhang, Z. He, et al, Machine-learning-based prediction methods for path loss and delay spread in air-to-ground millimeter-wave channels, IET Microwaves, Antennas & Propagation, Vol. 13, no. 8, pp. 1113–1121, July 2019.

Response 4: The last version of the manuscript did not contemplate a glimpse of A2G models. Therefore, the most recent version contains an additional subsection, 2.4.1 The A2G Channel Modeling, which embraces two novel classifications: Machine Learning and Geometry-based schemes. There is not literature enough that stands out the use of the division mentioned previously for developing a deep study on the A2G channel technical model. To this end, Table 4 covers eight references, which in turn includes the proposed literature among other related.

  • Point 5: It is recommended to add the SPC and LEO methods to the outdoor scene to make Fig.3 more complete.

Response 5: Consequently, Figure 3 was modified to fulfill the suggestions. In so doing, the height of the platforms was stated to envision the impact on the countryside and the advantages that bring to overcome the connectivity gap.

Reviewer 3 Report

The paper provides a high-level overview of coverage issues and potential solutions for rural not-spots. The paper covers a wide range of topics and, while generally well written, seems sometimes a bit unstructured. The paper would benefit if the content would be focused on fewer topics and those topics being presented in a more straightforward way.

There is little to none technical contribution in this paper and I believe this is also not the intention of the paper. As such, it is more suited as a background paper to develop future strategies and allocate funding for research and development.

While the language overall is good, some words seem to have an unusual use (while not being incorrect). Sometimes a word is missing, phrases such as "century XX" may refer to the 20th century but I doubt that this is always the intention (21st century?). Lines 272-285 are a duplication of 241-284 and should be removed. I did not explicitly check the paper for grammatical issues but would highly recommend proof-reading/editing by a native English speaker.

Author Response

Dear Reviewer,

In the below lines, I present the point-to-point response to each suggestion besides expressing my sincere gratitude for all the provided comments.

  • Point 1: The paper provides a high-level overview of coverage issues and potential solutions for rural not-spots. The paper covers a wide range of topics and, while generally well written, seems sometimes a bit unstructured. The paper would benefit if the content would be focused on fewer topics and those topics being presented in a more straightforward way.

Response 1: Accordingly, the article attempts to furnish the most suitable strategies to overcome the connectivity gap, especially in Latin America. The upsurge of non-conventional technologies—such as UAVs—to cover rural zones is the cornerstone of the article and the authors' aim for addressing novel research fields into broader Wireless Communication investigation. 

  • Point 2: There is little to none technical contribution in this paper and I believe this is also not the intention of the paper. As such, it is more suited as a background paper to develop future strategies and allocate funding for research and development.

Response 2: The last version of the manuscript did not contemplate a glimpse of A2G models. Therefore, the most recent version contains an additional subsection, 2.4.1 The A2G Channel Modeling, which embraces two novel classifications: Machine Learning and Geometry-based schemes. There is not literature enough that stands out the use of the division mentioned previously for developing a deep study on the A2G channel technical model. To this end, Table 4 covers eight references that approach the lack of technical framework, even though there is not the main aim of the article as you have noticed up.

  • Point 3: While the language overall is good, some words seem to have an unusual use (while not being incorrect). Sometimes a word is missing, phrases such as "century XX" may refer to the 20th century but I doubt that this is always the intention (21st century?).

Response 3: The use of unconventional words was restricted to allow the lectors an effortless understanding. 

  • Point 4: Lines 272-285 are a duplication of 241-284 and should be removed. I did not explicitly check the paper for grammatical issues but would highly recommend proofreading/editing by a native English speaker.

Response 4: The performed mistakes were thoroughly checked up, in the readiness of the corrected version of the manuscript.

Round 2

Reviewer 2 Report

This paper comprehensively investigates the use of three platforms to deliver broadband services. It also indicates that UAVs are considered a noteworthy solution to aboard the rural zones. Most of comments have been addressed in the manuscript. However, some small mistakes can be made before it’s accepted, such as Section 2.4 appears after the title of chapter 3.

Author Response

Dear Reviewer,

In the below lines, I present the point-to-point response to each suggestion besides expressing my sincere gratitude for all the provided comments.

  • Point 1: Most of comments have been addressed in the manuscript. However, some small mistakes can be made before it’s accepted, such as Section 2.4 appears after the title of chapter 3.

Response 1: The suggested mistakes were thoroughly checked up, in the readiness of the corrected version of the manuscript. Other changes regarding its presentation have been performed as well.